# Functional Characterization of *TkSRPP* Promoter in Response to Hormones and Wounding Stress in Transgenic Tobacco

**DOI:** 10.3390/plants12020252

**Published:** 2023-01-05

**Authors:** Gaoquan Dong, Mengwei Fan, Hainan Wang, Yadong Leng, Junting Sun, Jun Huang, Hao Zhang, Jie Yan

**Affiliations:** 1College of Life Sciences, Shihezi University, Shihezi 832003, China; 2Institute of Gardening and Greening, Xinjiang Academy of Forestry Sciences, Urumqi 830000, China

**Keywords:** abiotic stress, deletion analysis, natural rubber, phytohormones, promoter, small rubber particle protein

## Abstract

*Taraxacum kok-saghyz* is a model species for studying natural rubber biosynthesis because its root can produce high-quality rubber. Small rubber particle protein (SRPP), a stress-related gene to multiple stress responses, involves in natural rubber biosynthesis. To investigate the transcriptional regulation of the *TkSRPP* promoter, the full-length promoter PR0 (2188 bp) and its four deletion derivatives, PR1 (1592 bp), PR2 (1274 bp), PR3 (934 bp), and PR4 (450 bp), were fused to β-glucuronidase (GUS) reporter gene and transformed into tobacco. The GUS tissue staining showed that the five promoters distinctly regulated GUS expression utilizing transient transformation of tobacco. The GUS activity driven by a PR0 promoter was detected in transgenic tobacco leaves, stem and roots, suggesting that the *TkSRPP* promoter was not tissue-specific. Deletion analyses in transgenic tobacco have demonstrated that the PR3 from −934 bp to −450 bp core region responded strongly to the hormones, methyl jasmonate (MeJA), abscisic acid (ABA), and salicylic acid (SA), and also to injury induction. The *TkSRPP* gene was highly expressed under hormones and wound-induced conditions. This study reveals the regulation pattern of the *SRPP* promoter, and provides valuable information for studying natural rubber biosynthesis under hormones and wounding stress.

## 1. Introduction

Small rubber particle protein (SRPP) is located on the rubber particle film surface [1,2,3] and forms a subunit of the rubber transferase (RT-ase) complex, which affects rubber chain elongation and rubber quality [4,5]. A previous finding showed that the *Parthenium argentatum* Gray SRPP protein (GHS) could enhance the rubber biosynthesis activity in vitro [6]. Overexpression of *TkSRPP3* in *Taraxacum kok-saghyz* (*T. kok-saghyz*) could improve the rubber content in roots. RNA interference in *TkSRPP3* gene expression resulted in decreased rubber content and rubber hydrocarbon molecular magnitude in the root [4]. In addition to the above, the reduction of *Taraxacum brevicorniculatum* (*T. brevicorniculatum*) *TbSRPP* expression through RNAi technology could affect the stability of rubber particles and reduce the dry rubber content by 40~50% [7], indicating that *SRPP* family genes are crucial in rubber biosynthesis.

*SRPP* also plays a vital role in abiotic stress tolerance. For example, as an *SRPP* homologue, CaSRP1 gene overexpression in response to water stress in Capsicum plants could enhance drought tolerance and growth of *Arabidopsis thaliana* [8]. Three *SRPP* homologues (*SRP1*, *SRP2*, and *SRP3*) were identified in the model plant *Arabidopsis thaliana*, and their expressions were induced in response to drought as well as high and low temperature conditions to some extent. Compared with the wild-type plants, the overexpressed *SRP* genes in *Arabidopsis* showed higher vegetative growth, reproductive growth, and better tolerance to drought stress [9]. However, in producing rubber plants, the study of *SRPP* genes mainly concentrated on rubber biosynthesis [10,11], and the functional role of its stress tolerance in transgenic research has not been reported.

Previous studies found that *SRPP* promoters had many *cis*-acting elements related to abiotic stresses. The latex-specific *TbSRPP* promoter was regulated in response to external environmental conditions like light, wounds, and cold [12]. The *Hevea brasiliensis* (*H. brasiliensis*) *SRPP* promoter was cloned and induced in response to methyl jasmonate (MeJA), abscisic acid (ABA), gibberellin (GA), cold, heat, and wounding stress [13]. Moreover, the *HbSRPP* promoter was activated via HbMYC2 protein that regulated jasmonic acid (JA) responsive gene expression [14]. A drought-induced HbNAC1 was found to bind to the *cis*-element CACG in the *HbSRPP* promoter to increase the expression of β-glucuronidase (GUS) gene [15]. JA and ET-induced HbMADS4 could also activate the *HbSRPP* promoter [16]. These results indicate that the SRPP promoter can respond to various abiotic stresses.

*T. kok-saghyz* is an ideal plant for rubber biosynthesis research, containing 10 *TkSRPP/REF* genes in the genome [17]. No functional analysis of the *TkSRPP* genes for their role in stress tolerance or *TkSRPP* promoter activation has been determined. In this study, the isolation and functional analysis of the 5′ flanking sequence (PR0) of the *TkSRPP* gene is described. The PR0 promoter activities were conducted with the help of a GUS reporter gene through a histochemical GUS assay in tobacco. Deletion promoters were performed with Agrobacterium-mediated genetic transformation to analyze tissue specificity and hormones and wound response in transgenic tobacco. The gene expression of the *TkSRPP* promoter was analyzed under hormones- and wound-induced conditions. By analyzing the regulation pattern of the *TkSRPP* promoter, we can better understand the molecular mechanism by which *TkSRPP* promotes rubber biosynthesis in response to hormones and wounding stress.

## 2. Results

### 2.1. The 5′ Flanking Sequence of TkSRPP Promoter Cloning and Sequence Analysis

Based on the previous reported transcriptome data of *T. kok-saghyz* treated with MeJA, the *TkSRPP* gene was found up-regulated by MeJA [18]. To evaluate the expression pattern of the *TkSRPP* gene, the 5′ flanking sequence of the *TkSRPP* promoter (Genbank: MZ190894) was cloned by PCR, and its length was 2188 bp at 5′ flanking region of the ATG start codon (Figure 1). Based on the above *TkSRPP* gene sequence, the nucleotide sequence alignment analysis showed a high similarity of 97% between the correct clone sequence and the genome sequence of this promoter (Appendix A). Moreover, the *TkSRPP* promoter sequence distributed with TATA-box, CAAT-box, and other typical core elements was endowed with the characteristics of typical plant promoters (Figure 1).

### 2.2. cis-Element Analysis of TkSRPP Promoter

The *TkSRPP* promoter sequence was predicted to identify important *cis*-regulatory elements using PLACE [19] and PlantCARE [20] databases. From the predicted results, several vital regulatory elements were detected in the *TkSRPP* promoter (Table 1, Figure 1). For instance, a large number of elements are involved in light response (I-Box, G-box, chs-CMA1a, Gap-box, GA-motif) and tissue-specific expression (GATABOX). Plant hormone response elements include these with *cis*-elements involved in MeJA response (CGTCA-motif), ABA response (EBOXBNNAPA, MYCCONSENSUSAT, DRE core, RYREPEATBNNAPA), IAA response (TGA-element), GA response (P-box, WRKY71OS), SA response (TCA-element), and CK response (ARR1AT). Besides, abiotic stress-related elements are also present and include drought-inducing elements (MBS, CBFHV, DRECRTCOREAT), a wound response element (WUN-motif), a heat shock element (CCAATBOX1), a low-temperature response element (LTRECOREATCOR15), and a defense-related element (MYB1LEPR) (Table 1, Figure 1).

### 2.3. Activity Analysis of Various SRPP Promoters

Based on the correct *TkSRPP* promoter sequence, deletion fragments of PR0, PR1, PR2, PR3, and PR4 were cloned with sizes of 2188, 1592, 1274, 934, and 450 bp, respectively (Figure 2 and Appendix A). The recombinant plasmid pCAMBIA1304-PR0/PR1/PR2/PR3/PR4-*GUS* was further identified by double enzyme (*Nco1* and *BamH1*) digestion, and the results were consistent with the expected fragment size (Appendix A), suggesting that the five recombinant vectors be successfully constructed for subsequent experiments.

To detect the activity of the PR0 promoter, the GUS histochemical staining was performed on tobacco leaves treated with *Agrobacterium* harboring PR0 to evaluate the staining degree. The leaves of wild-type tobacco (negative control groups) were not stained while the tobacco leaf with 35S::*GUS* (positive control groups) was obviously stained to appear as a large blue patch. The experiment showed that the stained degree of tobacco leaf with PR0::*GUS* was deeper than that of the negative control groups but shallower than that of the 35S promoter (Figure 3), indicating that the transcriptional activity of the PR0 promoter was activated by upstream protein regulators in the tobacco leaf. Furthermore, the GUS activity of other deletion promoters (PR1/PR2/PR3/PR4) was also analyzed with GUS histochemical staining. The PR1, PR2, and PR3 promoter constructs were strongly stained compared with the negative control groups, whereas the PR4 construct was slightly stained in tobacco leaves (Figure 3). These results indicated that five promoters (PR0/PR1/PR2/PR3/PR4), driven by GUS reporter gene, showed distinct transcriptional activity due to promoter lengths.

### 2.4. Genetic Transformation of Different TkSRPP Promoters

To obtain transgenic tobacco plants with the promoter, pCAMBIA1304-PR0/PR1/PR2/PR3/PR4-*GUS* was transferred into tobacco by the leaf disk method. The callus of the infected tobacco leaves were differentiated into fascicled buds (Appendix A), then transferred to the rooting medium to grow into large plants (Appendix A), and finally transplanted to the nutrient soil (Appendix A). A pair of Kan gene primers and the PR4 promoter primers, respectively, were used to identify transgenic tobacco plants with different *SRPP* promoters, as partly shown in the picture (Appendix A).

### 2.5. Expression Analysis of SRPP Promoter in Tobacco Different Tissues

The histochemical promoter-GUS assay revealed that *AtSRP3* can be highly expressed in the *Arabidopsis* plants [9]. The rubber tissue-specific *HbSRPP* promoter in *T. brevicorniculatum* induced greater *GUS* gene expression in roots compared with the leaves tissues [12]. Therefore, the tissue specificity of the *TkSRPP* promoter deserved attention. The whole PR0 transgenic tobacco plant grown for half a month was stained by histochemical staining. It was found that the leaves tissues of PR0 transgenic tobacco were strongly stained compared with the wild-type tobacco, followed by the roots and stems (Figure 4a,b). Moreover, the *GUS* expression of roots and leaves tissues were approximately 2.73 and 3.73 times that in stems in the PR0 transgenic tobacco, respectively (Figure 4c). These results showed that the *TkSRPP* promoter, mainly in the tobacco leaf and root, performed obvious transcriptional activity displayed by the *GUS* reporter gene.

### 2.6. GUS Activity Analysis of TkSRPP Promoters under Hormones and Wounding Stress 

Under untreated conditions, the GUS activity of PR0, PR1, PR2, and PR3 (11.01, 9.49, 8.20, and 2.21-fold, respectively; *p* < 0.01) was significantly higher than that of PR4 in transgenic tobacco (Control in Figure 5). In particular, when the *TkSRPP* promoter was truncated to the PR3 promoter, the regulated GUS activity decreased significantly (Figure 5).

To ascertain the distribution of stress-related *cis*-elements in the *TkSRPP* promoter, the transgenic tobacco with deletion *TkSRPP* promoters were treated with ABA, MeJA, SA hormones, and wounding stress. After ABA treatment, the GUS activity significantly increased for the PR0, PR1, PR2, and PR3 (2.08, 2.76, 2.47, and 3.48-fold, respectively; *p* < 0.01), and PR4 (1.50-fold; *p* < 0.05) promoters compared with the control (untreated conditions) (Figure 5a), suggesting the key *cis*-elements that respond to ABA resided in the −1284 to −1 bp region of the promoter. After treatment with MeJA, the GUS activity of the PR0, PR1, PR2, and PR3 promoters significantly increased (1.52, 2.31, 4.84, and 7.53-fold, respectively; *p* < 0.01) compared with the control, while no significant changes were detected for the PR4 promoter (Figure 5b). It may be that the promoter region from −2188 to −405 bp responds to MeJA positively. After SA treatment, GUS activity increased for PR1 and PR2 (1.69 and 1.88-fold; *p* < 0.01) compared with the control, while the GUS activity had no significant effect on PR0, PR3, and PR4 (Figure 5c). The results showed that the promoter region from −1592 to −934 bp containing SA-related *cis*-elements responds to SA. Compared with control, the GUS activity increased more for PR3 (2.33-fold; *p* < 0.01) under wound stress, whereas the GUS activity of other promoters (PR0, PR1, PR2, and PR4) did not show significant change (Figure 5d). The promoter region from −934 bp to −450 bp harboring *cis*-elements appears to respond strongly to the wound. Interestingly, the PR3 promoter can be induced by ABA, MeJA, and wounding stress (Figure 5), indicating the fragment from −934 bp to −450 bp was the key region of the *TkSRPP* promoter responses to stress. Collectively, these results demonstrated that the *TkSRPP* promoter has a biological function involved in ABA, SA, MeJA and wounding responses.

### 2.7. Expression Analysis of TkSRPP Gene under Hormones and Wounding Stress

Due to the *cis*-acting element composition effects on the promoter playing an important role in determining the transcription level and function of the gene, qRT-PCR experiments were carried out to analyze the gene expression of this *TkSRPP* promoter in response to various stresses. The observation showed that the expression levels of the *TkSRPP* gene in *T. kok-saghyz* were up-regulated (9.10, 14.59, respectively; *p* < 0.01) by ABA, MeJA treatment for six hours, consistent with the results of GUS activity of PR0 under above stress conditions. Moreover, compared with the control, the *TkSRPP* gene expression was induced to rise to 5.33 and 2.03-fold, respectively; *p* < 0.01 under SA and wound treatment for six hours (Figure 6); this was not consistent with the results of GUS activity of PR0 under SA and wounding stress conditions attributed to incomplete flanking sequences of the pre-cloned *TkSRPP* gene. In short, the results indicated that the *TkSRPP* gene might play a vital role in hormones and wounding stress responses.

## 3. Discussion

The involvement of *SRPP* in the natural rubber biosynthesis has been found [10]. However, the role and function of the *SRPP* gene in abiotic stress remain inadequately studied. In this study, the 5′ flanking sequence of the *TkSRPP* promoter was cloned, and it predicted several *cis*-regulatory elements associated with signal molecules and environmental stresses. To investigate the regulation mechanism of *TkSRPP* expression, deletion analysis of the *TkSRPP* promoter in transgenic tobacco was carried out for the critical promoter regions responding to hormones and wounding stress. Moreover, the expression of the *TkSRPP* gene was enhanced under ABA, MeJA, SA, and wounding stress.

### 3.1. The Key Region Function of SRPP Promoter

The interaction of transcription factors with *cis*-acting elements is essential in the regulation of plant gene expression responding to hormones and environmental stimulus [21]. Data showed that the GUS activity of deletion *TkSRPP* promoters was significantly increased after ABA treatment (Figure 5a). Moreover, the ABA response elements (EBOXBNNAPA, MYCCONSENSUSAT, DRE core, RYREPEATBNNAPA) were also found in the *TkSRPP* promoter (Figure 1), among which EBOXBNNAPA elements were widely distributed and contained an E-box motif (CANNTG) that acts as a binding site of *bHLH* transcription factors in response to ABA, such as the *CmLOX08*, *GmRD26* and *TPS21* promoter [22,23,24,25]. Therefore, the *TkSRPP* promoter region may have a large number of ABA-related *cis*-elements that respond positively to ABA treatment. The *TkSRPP* promoters (PR0, PR1, PR2, and PR3) were upregulated by the MeJA treatment (Figure 5b), suggesting that the MeJA-responsive elements were widely distributed within this region (from −2188 bp to −450 bp). However, the predictive analysis of the *TkSRPP* promoter only found one TGACG-motif (TGACG) (Figure 2b), a *cis*-acting regulatory element in the MeJA responsiveness [26,27]. There might be other potential MeJA-inducible elements that were not predicted through the *TkSRPP* promoter. The SA treatment led to a significant increase for GUS expression of the PR1 and PR2 promoters (Figure 5c). Moreover, the presence of a TCA-box motif in this promoter region ranged from PR2 to PR3 (Figure 1) is a *cis*-acting element in SA responsiveness [28]. Thus, we speculated that the upstream sequence of the PR1 promoter may have a repressive element of SA response. The role played by TCA motif in regulating the expression of *TkSRPP* is similar to the role of SA as a signal molecule in *VpTNL1* from *Vitis pseudoreticulata* [29]. The *HbSRPP* promoter with a wound response element can regulate GUS activity through wound treatment [13]. The GUS activity of the PR3 promoter was regulated by the wound (Figure 5d), indicating the existence of wound-inducible elements in the region from −934 to −450 bp, consistent with the results of promoter prediction (Figure 1 and Figure 2). Differences in induction intensity of various *TkSRPP* promoters under various hormones and wounding stress may be related to the distribution of specific *cis*-elements in their sequences. Although the sequence of the *cis*-acting element cannot be precisely identified, the *TkSRPP* promoter does harbor fruitful *cis*-acting elements associated with ABA, MeJA, SA, and wounding stress.

### 3.2. SRPP Is a Typical Stress-Induced Gene in Plants

The functions of homologous family genes seem to be similar in the growth and development of plants and the process of stress responses [30]. The *SRPP* family genes can promote the biosynthesis of natural rubber, such as *T. kok-saghyz*, *T. brevicorniculatum*, and *H. brasiliensis* [2,4,7,31]. Moreover, the *SRPP* family genes could respond diversely to ethylene stimulation among different rubber tree clones [32]. Young *Arabidopsis thaliana* plants overexpressing *TbSRPP2* and *TbSRPP3* were endowed with higher tolerance to drought stress than wild-type plants [31]. Deletion analysis revealed that *HbSRPP* and *TbSRPP* promoters could respond to hormones and abiotic stresses [12,13]. In this study, *TkSRPP* promoters were predicted to have a large number of *cis*-elements associated with the stress response (Figure 1) and can be regulated by ABA, MeJA, SA, and injury in transgenic tobacco (Figure 5). The *TkSRPP* gene has high expression when induced by ABA, MeJA, SA, and wound stresses (Figure 6). 

Although only one abiotic stress (wound) was performed in this study, the predicted *TkSRPP* promoter found several drought-related *cis*-elements (MBS, CBFHV, DRECRTCOREAT) and other stress responsive elements (Figure 1). We thus speculated that the *TkSRPP* promoter has the same function as the *HbSRPP* and the *TbSRPP* promoter involved in the biological process of a variety of hormones and abiotic stress. In addition, *Capsicum annuum* and *Arabidopsis*, the non-rubber-producing plants, also have *SRPP* homologous genes, but these genes (*AtSRPs*, *GaSRP*) might participate in the stress response, such as salt and drought resistance [8,9]. In short, the homologous *SRPP* genes in rubber-producing plants and non-rubber-producing plants are a class of stress-inducing genes, but whether they have a similar function in rubber synthesis needs further study.

### 3.3. The TkSRPP Gene Is Expected to Promote Rubber Biosynthesis under Hormones and Wounding Stress

JA is a broad-spectrum plant hormone that is essential for the regulation in plant secondary metabolism [33]. Studies have shown that external JA application could promote rubber biosynthesis and laticifer cell differentiation [14,34]. For instance, the expression of the *SRPP* gene involved in rubber biosynthesis was up-regulated by MeJA induction in the transcriptome of *H. brasiliensis* and *T. kok-saghyz* [18,35]. Meanwhile, some transcription factors that respond to JA or MeJA treatments might regulate *SRPP* gene expression, such as HbMYC2, HbMADS4, HbWRKY83, etc. [14,16]. In this study, the *TkSRPP* promoter region (from 1256 to 456 bp) could enhance GUS activity induced by exogenous MeJA (Figure 5). Moreover, the results of qRT-PCR showed that the *TkSRPP* gene expression could also be up-regulated by MeJA treatments (Figure 6) and the *TkSRPP* gene showed high expression in RNA-seq data of latex tissue [17]. Thus, the *TkSRPP* gene can be identified as a key candidate gene to study the regulation of rubber biosynthesis through JA signaling pathways.

JA performs a critical role in regulating plant responses to wounding stress [36]. Previous findings revealed that JA-mediated wound signaling promotes plant regeneration [37,38]. Compared with control, the expression of *TkSRPP* gene was up-regulated (2.03-fold) under wound treatment for 6 h, but this was obviously lower than MeJA treatment (Figure 6). Moreover, the GUS activity could reflect that the *TkSRPP* promoter is more easily regulated by MeJA than the wound (Figure 5). This may be because wounding stress promotes endogenous JA synthesis and then regulates the expression of the *TkSRPP* gene by activating JA signaling pathway. The predicted *TkSRPP* promoter found a wound-responsive element in the region from −934 bp to −450 bp (Figure 2b), and the GUS activity of PR3 promoter had significantly higher than that of control under wounding stress, while other deletion promoters showed no significant change (Figure 5d). The above results indicated that the induction of wounding stress on the regulation of *TkSRPP* promoter is a complex biological process, which needs to be further studied.

More strong evidence suggests that plant responses to various stresses depend on the crosstalk among hormonal signaling pathways, rather than on the single role of each one [39]. It is exemplified by case that the synergistic effect of both JA and ABA signaling suppressed seed germination of *Arabidopsis* [40]. The transcription factor *ANAC032* from *Arabidopsis* was characterized by dual roles that directly repress JA signaling and activate SA signaling under the *Pseudomonnas syringane* infection [41]. Here, we noticed that the GUS activity in the deletion analysis showed similar expression patterns treated by ABA and MeJA (Figure 5), suggesting that both may play a synergistic role in the regulation of the *TkSRPP* promoter. On the contrary, under SA treatment conditions (Figure 5), the GUS activity of only two deletion promoters (PR1 and PR2) had significantly higher than that of the control. However, the GUS activity of the PR0 promoter did not change, which may be due to the existence of JA response elements in the PR0-PR1 region inhibiting the regulation of SA. The crosstalk between the three hormones (ABA, SA, and MeJA) co-regulates the expression of the *TkSRPP* gene involved in rubber biosynthesis and the mechanism and reason for this need to be further studied. Collectively, the regulation pattern of the *SRPP* promoter in response to SA, ABA, MeJA and wounding stress were revealed, which provides a theoretical basis for us to better understand how these conditions are involved in rubber biosynthesis process.

## 4. Materials and Methods

### 4.1. Plant Materials, Growth Conditions

The leaves tissue samples were obtained from *T. kok-saghyz* growing in the culture room at 21 ± 2 °C, under a 16/8 h day/night photoperiod for two months. The *Nicotiana benthamiana* plants for agro-infiltration were cultivated in a culture room at 25 °C under a 16/8 h day/night cycle for six weeks. The tobacco (*Nicotiana tabacum*) was grown in a sterile bottle at 25 °C under a 16/8 h day/night photoperiod. 

### 4.2. Cloning of the HbSRPP Genomic Sequence

The 5′ flanking sequence of the *SRPP* promoter was obtained from the *T. kok-saghyz* genome of the Genome Warehouse (GWH; http://bigd.big.ac.cn/gwh/, accessed on 10 March 2021) under the accession number PRJCA000437 [17], based on sequence alignment of the *TkSRPP* gene using Blast 2.11.0 software (ftp://ftp.ncbi.nlm.nih.gov/blast/executables/blast+/LATEST/, accessed on 3 January 2021). Genomic DNA was isolated from *T. kok-saghyz* leaves as per the method proposed by [18]. The primers (Appendix A) for cloning of the promoter region were designed using Primer 5.0 software. PCR experiment was performed using the total DNA of leaf as a template, and the DNA polymerase is PrimeSTAR Max DNA Polymerase (TAKARA, Beijing, China). The PCR program was as follows: 30 cycles of 10 s at 95 °C, 20 s at 55 °C, and 2.5 min at 72 °C, with a final extension at 72 °C for 8 min. PCR products were detected by 1% agarose gel electrophoresis and recovered with the agarose gel DNA recovery kit (TIANGEN, Beijing, China). The recovered product was connected with the pMD-19T vector (TAKARA, Beijing, China) at 16 °C for 8 h according to the manufacturer’s instructions. The recombinant plasmid was then transformed into *E. coli* DH5α strain, and several positive single colonies were selected and sequenced for confirmation.

### 4.3. Analysis of cis-Regulatory Elements in the TkSRPP Promoter

The sequence of the *TkSRPP* promoter was predicted to identify the existing *cis*-elements of the *TkSRPP* promoter with PlantCARE (http://bioinformatics.psb.ugent.be/webtools/plantcare/html/, accessed on 12 March 2021) and PLACE (http://www.dna.affrc.go.jp, accessed on 16 March 2021) database [19,20].

### 4.4. Construction of the TkSRPP Promoter::GUS Plasmids

First, for construction of the *TkSRPP* promoter driving *GUS* fusion genes, the 5′ flanking sequence (−2188 bp) of *SRPP* gene and four deletion fragments encompassing different lengths of the *TkSRPP* promoter (−1592 bp, −1274 bp, −934 bp, and −450 bp to −1 bp, respectively) were obtained by PCR amplification using five sets of specific PCR primers with *BamHI* and *NcoI* sites (Appendix A). These fragments’ amplification was performed using pMD-19T-*PTkSRPP* plasmid as a template to obtain their sequence fragments. Then, five fragments were respectively cloned into the pMD-19T vector. The recombinant plasmids of 19T vectors were digested by *BamHI* and *NcoI*, and five fragments were connected with the pCAMBIA1304-35S::GUS vector (Figure 2) using T4 DNA Ligase (TAKARA, Beijing, China). Finally, all of the structures were transformed into the *E. coli* DH5α strain, and positive clones were verified by sequencing, colony PCR, and enzyme digestion reaction. 

### 4.5. Histochemical Straining Assays

Histochemical GUS assays were performed as described elsewhere [18]. The suspension (1 mL) of *Agrobacterium tumefaciens* strain (GV3101) harboring a *TkSRPP* promoter plasmid was inoculated into 50 mL LB liquid medium supplemented with 70 mg/L rifampicin (rif) and 50 mg/L kanamycin (Kan) at 28 °C with shaking until OD600 value was 0.8. *Agrobacterium* cells were harvested after centrifugation at 5000× *g* for 5 min, suspended in infiltration solution (10 mM MES, pH 5.7, 10 mM MgCl2, and 20 µM AS), incubated at 28 °C for 3 h at dark. The final bacterial suspension was injected into tobacco lower epidermal leaf via a needleless syringe and the whole tobacco leaf was infused with *Agrobacterium*. The tobacco plants were then cultivated in a constant temperature incubator under a 16/8 h day/night cycle at 25 °C for 48 h. The treated tobacco leaves were prepared for subsequent GUS staining experiments.

Histochemical staining was performed following the procedures described by Jefferson [22]. The tobacco leaves with *Agrobacterium*-mediated transient transformation were incubated in GUS staining solutions, including 80 mM sodium phosphate buffer (pH 7.0), 0.5 mM potassium ferricyanide, 0.5 mM potassium ferrocyanide, 10 mM EDTA, 1 mg/mL 5-Bromo-4chloro-indolyl-b-D-glucuronide (X-Gluc), and 0.1% Triton X-100 for 24 h at 37 °C. After staining, the tissues were transferred to 70% ethanol to decolorize at 37 °C for about two days. Stained tissues were examined and taken photos with a scanner.

### 4.6. Transgenic Plant Generation 

Various truncation promoters were transformed into tobacco by stable genetic transformation to verify the functions of deletion promoters. The sterile tobacco leaves were cut into 1 × 1 cm area segments with scissors and then placed in the Murashige and Skoog (MS) medium in the dark for two days. Of the recombinant plasmids carried by the *Agrobacterium tumefaciens* (GV3101), five were transformed into tobacco by the leaf disc method [42]. Transgenic tobacco plants were screened using Kan (100 mg/L) and identified by a pair of primers of the *kan* gene and the *TkSRPP* promoter primers (Appendix A). 

### 4.7. Tissue Specificity of TkSRPP Promoter 

The T2 transgenic tobacco with the *TkSRPP* promoter and the wild-type tobacco plants were grown on MS in sterile culture bottles. The GUS staining experiments were conducted to analyze the tissue-specific functions of the *TkSRPP* promoters in transgenic tobacco. The GUS expression of different tissues (leaf, stem, and root) of the *TkSRPP* transgenic tobacco was also performed by qRT-PCR to assess the tissue specificity of the *TkSRPP* promoter. 

### 4.8. Abiotic Stress Treatments

The T2 transgenic tobacco plants were transplanted into flowerpots and grown for two months, under a 16/8 h day/night photoperiod. The leaves of transgenic tobacco were sprayed with 100 µM abscisic acid (ABA), 1 mM salicylic acid (SA), and 1 mM methyl jasmonate (MeJA), respectively, with deionized water treatment as control groups. The injury stress used scissors to cut the transgenic tobacco leaves into strip wounds with a distance between them of 2 cm. The transgenic tobacco plants were used as control groups under normal conditions. Three biological replicates samples were collected after six hours and were used to measure the GUS activity. In addition, the three months wild-type *T. kok-saghyz* plants were treated with ABA, SA, MeJA and wound for six hours in the same way as above. The leaf tissues of three biological duplicates were collected for liquid nitrogen quick-freezing and stored in the −80 °C refrigerator.

### 4.9. GUS Fluorimetric Assays

The Stable expression of GUS activity in the treated tobacco leaves was measured as described elsewhere. The tobacco leaf tissue (0.5 g) was homogenized in 1 mL extraction buffer [50 mM sodium phosphate buffer (pH 7.0), 10 mM EDTA (pH 8.0), 0.1% Triton X-100, 0.1% (*w/v*) sodium dodecyl sulfate, 10 mM b-mercaptoethanol]. The supernatant (50 μL) centrifuged at 13,000 rpm for 15 min at 4 °C was collected and mixed with 390 mL pre-warmed (37 °C) *GUS* assay solution (1 mM methyl-4-umbelliferyl-D-glucuronide in extraction buffer) and incubated at 37 °C to collect the liquid sample (100 μL) every 30 min. The liquid samples were added to the (900 μL) stop buffer (0.2 M Na_2_CO_3_). The fluorescence signal was measured with an ultraviolet and visible spectrophotometer (f97xp; lengguang tech., Shanghai, China) at excitation and emission wavelengths of 365 nm and 455 nm. Stop buffer and 0 nM to 100 nM were used for 4-methylumbelliferone (4-MU) calibration and standardization. The total protein concentration was measured with bovine serum albumin as a standard, following the methods of the previous report [43]. The GUS activity was determined by nM of 4-MU generated per min per mg soluble protein and performed with three duplicates.

### 4.10. qRT-PCR

The total RNA was extracted with an EasyPure^®^ Plant RNA Kit (TRAN, Beijing, China). The first strand of cDNA was synthesized with the EasyScript^®^ One-Step gDNA Removal Kit and cDNA Synthesis Supermix (TRAN, Beijing, China) following the instructions of manufacturers. Meanwhile, Primer 5.0 software was used to design the primers (Appendix A) for qRT-PCR. The qRT-PCR reaction was performed on the Roche 480 platform with LightCycler 480 SYBR Green Master Mix (Roche, Shanghai, China). The qRT-PCR procedure was performed using the following profile: preincubated at 95 °C for 5 min, followed by 40 cycles of denaturation at 95 °C for 10 s, annealed at 58 °C for 10 s, and extended at 72 °C for 30 s. The *TkGAPDH* [18] and *NbEF1* [44], respectively, were used as reference genes. The full experiments were performed with three biological and experimental duplicates to analyze quantitative data with the 2^−ΔΔct^ methods [45].

### 4.11. Statistical Analysis

The data of three independent experiments were analyzed with a one-way analysis of variance. The value *p* ≤ 0.05 was considered significant and charts were generated with GraphPad Prism 8 software.

## 5. Conclusions

In this study, we isolated and characterized the 5′ flanking sequence of the *TkSRPP* promoter associated with many abiotic response elements. Based on the performance of the transgenic tobacco plants, we concluded that the *TkSRPP* promoter plays a vital role in responding to SA, MeJA, ABA, and wounding stress. Moreover, the results of its gene expression also confirmed the above conclusion. These results enhance our understanding of the role of the *TkSRPP* promoter in the regulation of SA, MeJA, ABA, and wounding responses, and provide useful information for further study of the rubber biosynthesis mechanisms by the *TkSRPP* promoter in response to hormones and wounding stress.

## Figures and Tables

**Figure 1 plants-12-00252-f001:**
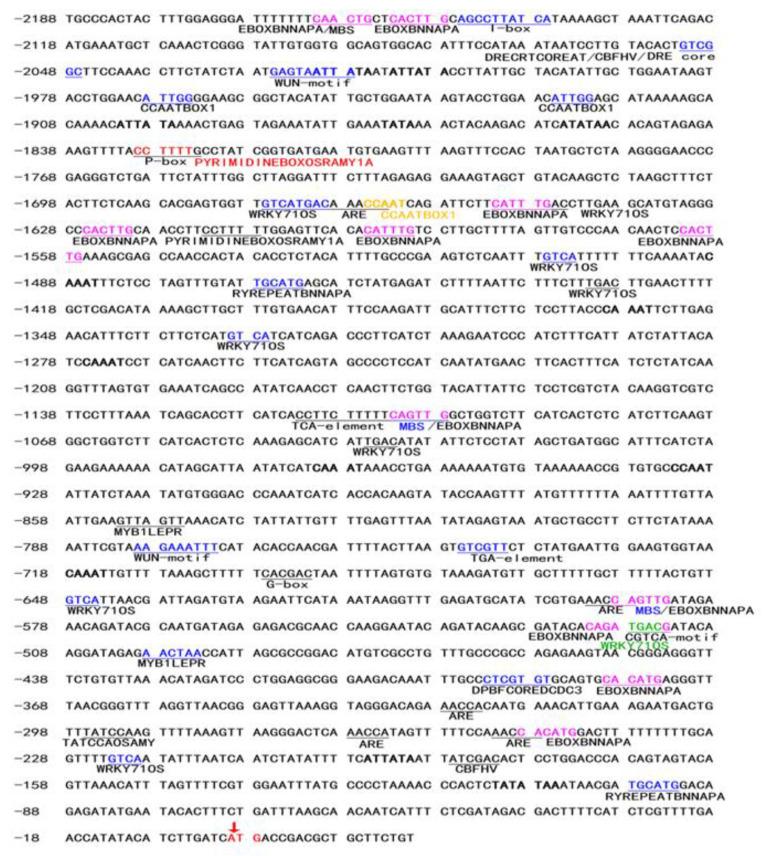
Physical map of the *TkSRPP* promoter. The “A” of the translation initiation code “ATG” of *TkSRPP* was designated as “+1”. The TATA-box is highlighted in bold. Putative *cis*-acting elements are underlined, colored and labeled. The *cis*-acting element sites on the negative strand and the double strands are shown in blue and pink, respectively. All the stress-responsive motives are represented by different colours.

**Figure 2 plants-12-00252-f002:**
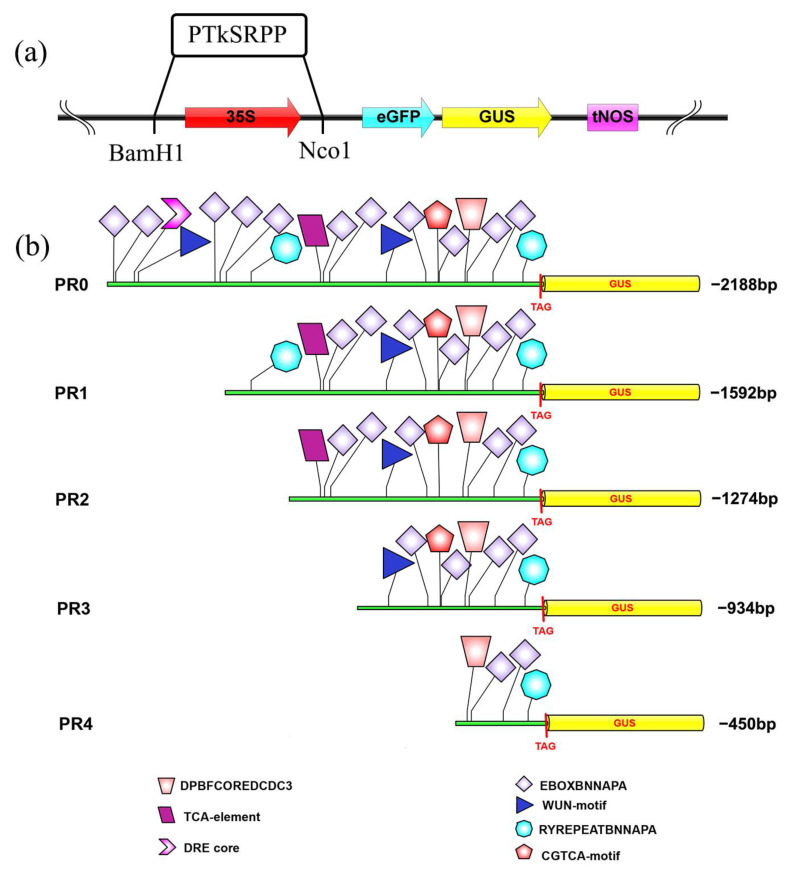
Schematic representation of the *TkSRPP* promoter regions controlling the expression of the GUS reporter gene. (**a**): Schematic drawing of the *TkSRPP* promoter expression cassette in the pCAMBIA1304::*GUS* expression vector. (**b**): Diagram of the main *cis*-acting elements in the full-length *TkSRPP* promoter (PR0) and four promoter modules (PR1, PR2, PR3 and PR4). The *cis*-elements associated with phytohormone, and abiotic stresses are represented by different coloured boxes.

**Figure 3 plants-12-00252-f003:**
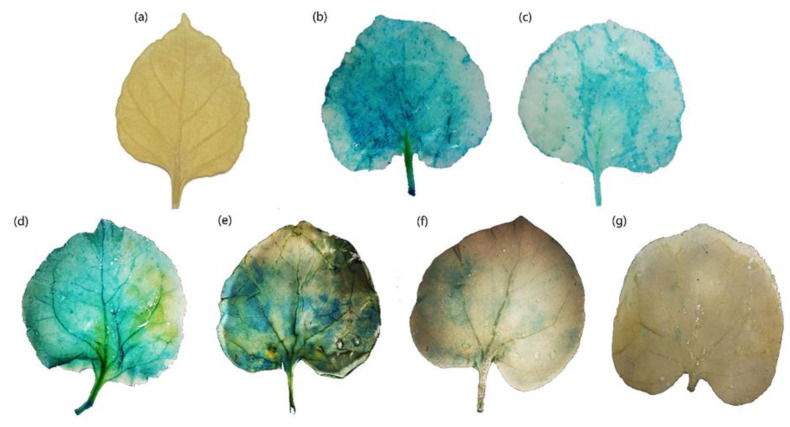
Activity analysis of different deletion constructs of *TkSRPP* promoter in tobacco plants. (**a**): Blank control; (**b**): CAMV 35S promoter regulates GUS expression; (**c**–**g**): the GUS activity was regulated by PR0, PR1, PR2, PR3 and PR4, respectively. The GUS histochemical staining of five deletion constructs was performed using transient transformation methods.

**Figure 4 plants-12-00252-f004:**
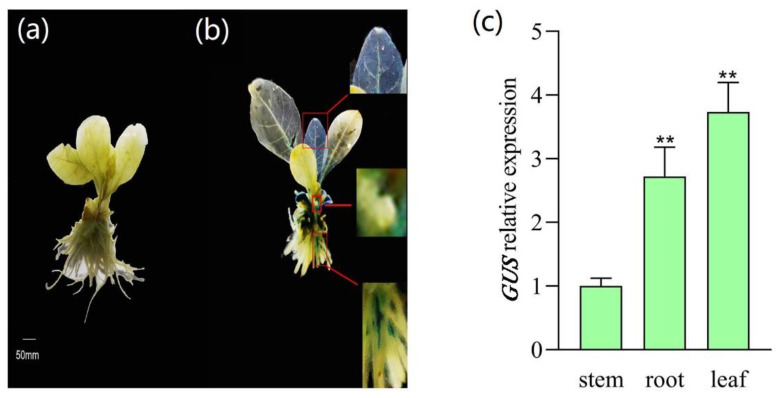
Tissue-specific analysis of the *SRPP* promoter in tobacco. (**a**): Wild-type tobacco served as control. (**b**): The GUS histochemical staining of the full-length *TkSRPP* promoter transgenic tobacco. The upper right, middle right, and lower right images show local magnification of leaves, stems and roots, respectively. (**c**): Relative expression of *GUS* in different tissues was evaluated by RT-PCR. The bars represent standard errors, and the asterisks indicate statistical significance determined by the student’s *t*-test (** *p* ≤ 0.01).

**Figure 5 plants-12-00252-f005:**
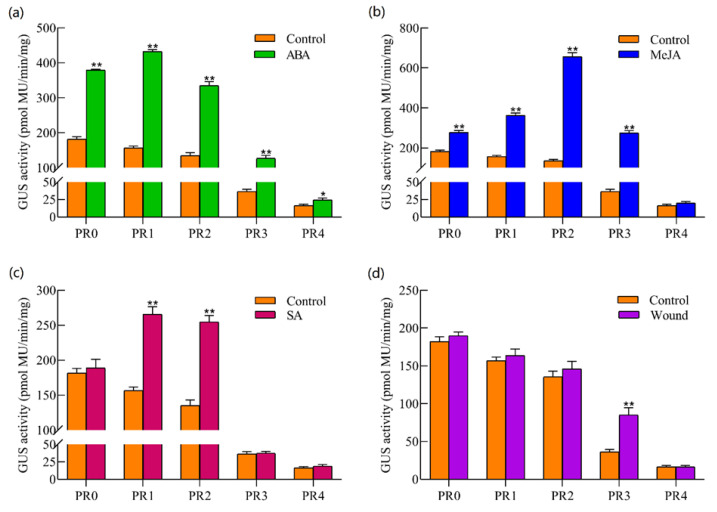
Assessment of stress and hormone responsiveness of the *TkSRPP* promoter and its deletion fragments. (**a**–**d**): The GUS activity analysis of five deletion promoters in response to abscisic acid (ABA), methyl jasmonate (MeJA), salicylic acid (SA) and wound treatment, respectively. The bars represent standard errors, and the asterisks indicate statistical significance determined by the student’s *t*-test (* *p* ≤ 0.05, ** *p* ≤ 0.01).

**Figure 6 plants-12-00252-f006:**
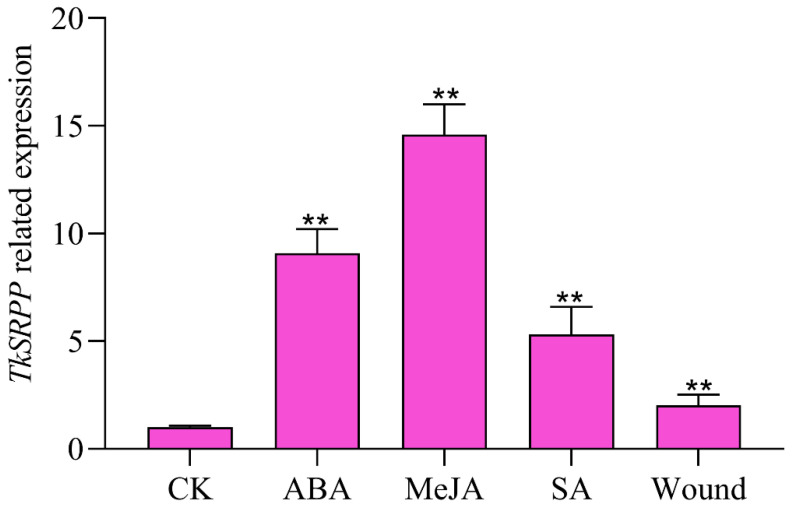
*TkSRPP* expression profile in *T. kok-saghyz* under multiple stresses. To evaluate the *TkSRPP* expression profile, *T. kok-saghyz* was treated by ABA, MeJA, SA and wound for six hours, and the *TkSRPP* expression was analyzed by qRT-PCR. GAPDH were used as reference genes. Compared with the control (CK), the SRPP relative expression of the treatment groups was calculated by the 2^−ΔΔCt^ method in biological triplicates (n = 3). The bars represent standard errors and the asterisks indicate statistical significance determined by the Student’s *t*-test (** *p* ≤ 0.01).

**Table 1 plants-12-00252-t001:** Biological analysis of the *cis*-acting element of the *TkSRPP3* promoter from *T. kok-saghyz*.

*cis*-Elements	Number	Function	Motif
CAAT-box	38	common *cis*-acting element in eukaryotic genes	CAAT
TATA-box	28	core promoter element around −30 of transcription start	TATA
ARE	5	*cis*-acting regulatory element essential for the anaerobic induction	AAACCA
MBS	3	MYB binding site involved in drought-inducibility	CAACTG
O2-site	3	*cis*-acting regulatory element involved in zein metabolism regulation	GTAC
GT1-motif	2	light responsive element	GGTTAA
TCA-element	4	*cis*-acting element involved in salicylic acid responsiveness	CCATCTTTTT
I-box	1	part of a light responsive element	GATAA
chs-CMA1a	1	part of a light responsive element	TCACTTGA
Box 4	1	part of a conserved DNA module involved in light responsiveness	ATTAAT
CAT-box	1	*cis*-acting regulatory element related to meristem expression	GCCACT
CGTCA-motif	1	*cis*-acting regulatory element involved in the MeJA-responsiveness	CGTCA
G-box	2	*cis*-acting regulatory element involved in light responsiveness	CACATGG
MRE	1	MYB binding site involved in light responsiveness	AACCTAA
TGACG-motif	1	*cis*-acting regulatory element involved in the MeJA-responsiveness	TGACG
WUN-motif	1	wound-responsive element	TCATTACGAA
Gap-box	1	part of a light responsive element	CAAATGAA(A/G)A
GA-motif	1	part of a light responsive element	ATAGATAA
GATA-motif	1	part of a light responsive element	AAGATAAGATT
P-box	1	gibberellin-responsive element	CCTTTTG
TCT-motif	1	part of a light responsive element	TCTTAC
ABRE	1	Abscisic acid response *cis*-acting element	ACTG
TGA-element	1	auxin-responsive element	AACGAC
3-AF1 4-binding site	1	light responsive element	TAAGAGAGGAA
W-box	1	WRKY binding wite involved in salicylic acid (SA)-induced responsiveness	TTGAC
TCA	1	*cis*-acting element involved in salicylic acid responsiveness	TCATCTTCAT
GATABOX	48	Required for light regulation and tissue-specific expression	GATA
EBOXBNNAPA	24	Abscisic acid response *cis*-acting element	CANNTG
MYBCORE	9	ABA response element	CNGTTR
CCAATBOX1	5	Work with HSE to increase the activity of the promoter	CCAAT
WBOXATNPR1	4	Important response component of SA	TTGAC
MYB	5	Unknown	CAACCA
MYC	4	Unknown	CATTTG
MYB-like	1	Unknown	TAACCA
Myb	4	Unknown	CAACTG
Myb-binding site	1	Unknown	CAACAG
STRE	4	Unknown	AGGGG

## Data Availability

The data presented in this study are available on request from the corresponding author.

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
