# Peer review of "Functional Characterization of TkSRPP Promoter in Response to Hormones and Wounding Stress in Transgenic Tobacco"

_plants, 2023, doi:10.3390/plants12020252_

Round 1

Reviewer 1 Report

In their paper, Dong et al studied the transcription regulation of the TkSRPP gene from Taraxacum kok-saghyz species by using different promoter regions and studying their expression in transgenic tobacco. In order to identify the possible involvement of TkSRPP gene to stresses, authors also studied the response of these transgenic tobacco lines to different abiotic stressors and wounding stress, presenting some interesting results. The whole manuscript The manuscript is well written, results are presented clearly and the conclusions are well supported by the results. However, there are some important points that need to be clarified. Most important points are:

Lines 165 – 168: GUS activity between untreated different promoter regions transgenic tobacco plants is missing from Figure 5.

Lines 187-189; according to Figure 5 and results, PR3 promoter is not statistically significantly induced by SA in comparison to the control. So, SA does not seem to be an inducer of this promoter. This phrase needs to be corrected.

Lines 236-237, 242-243, 246-248, 269-270, 271-273, 300-301, 303-306, 317-320.

Authors should pay attention to the choice of words/syntax/grammar in these phrases, they do not make sense as they are.  

How many T2 transgenic tobacco plants were treated with stressors and what does the biological replicate consist of? Similarly, how many 3-months WT plants were treated with the stressors and what does the biological replicate duplicate consist of? In other words, how many leaves from how many plants consist one biological replicate? In Lines 441-443 authors refer to three independent experiments. What does that mean?

Lines 287-290; check references format.

Reviewer 2 Report

The article presents interesting and original results. Authors cloned TkSRPP promoter region using primers based on available genome sequencing results. Authors prepared several deletion mutants and tested their activity in different organs and in response to different plant growth regulators. Study is well planned and performed, conclusions are supported by obtained results. Following minor comments should be addressed before the publication:

Line 32; correct the lack of space.

Cis-active or cis-elements, write cis in italics, check the whole text and descriptions to figures.

In the introduction section Authors should provide the sequences of cis-active elements responding to MeJA, SA, ABA, wounding and provide appropriate citations.

Line 155: correct the sentence „tissues were 2.73 and 3.73 times that....” to „tissues were approximately 2.73 and 3.73 times that.…”

Section 4.2 Provide name of DNA polymerase used to amplify the promoter region.

Line 270 should be Kan 50 mg/L not Kan 50 mg/mL- the last value is 1000x too high concentration. The better method is writing 50 mg L-1 not 50 mg/mL . Correct in the entire text.

Section 4.8 Provide the time of plant regulator action (6 hrs) in other words the duration of exposition to MeJa, SA, ABA etc. Although it is provided in materials and methods section, addition of  this information here will result in more clear text composition.

Line 4.10 how the quality of RNA was assessed? Provide the amount of RNA per sample. Provide the length of RT-PCR product for reference and tested gene.

Fig 3; write TkSRPP in italics

Fig 4 write TkSRPP not SRPP

Figure 4c: Correct the sentence :”Relative expression of GUS in different tissues”  to following sentence: (c): Relative expression of GUS in different tissues was evaluated by RT-PCR. Write GUS in italics. In Fig 4c write also GUS in italics (0Y axis) .

Fig 6; Description of Fig 6. TkSRPP write in italics: TkSRPP. Write also in the Fig 6 theTkSRPP in italics (0Y axis).

Discussion section

Authors correctly studied cis-active elements in the entire promoter region. However, cis-elements localized in the proximal promoter region , usually within 200-300 bp from transcription start site are more biologically relevant. Compare the following works (Mironova et al. 2014; Keilwagen et al. 2011; Yu et al. 2016) and add 1-2 sentences to describe shortly this fact. Sample references below:

Mironova VV, Omelyanchuk NA, Wiebe DS, Levitsky VG (2014) Computational analysis of auxin responsive elements in the Arabi[1]dopsis thaliana L. genome. BMC Genomics 15:S4

Keilwagen J, Grau J, Paponov IA, Posch S, Strickert M, Grosse I (2011) De-novo discovery of diferentially abundant transcription factor binding sites including their positional preference. PLoS Comput Biol 7:e1001070. https://doi.org/10.1371/journal.pcbi.1001070

Yu ChP, Lin JJ, Li WH (2016) Positional distribution of transcription factor binding sites in Arabidopsis thaliana. Sci Rep 6:25164
